# Immunization with Neural-Derived Peptides in Neurodegenerative Diseases: A Narrative Review

**DOI:** 10.3390/biomedicines11030919

**Published:** 2023-03-16

**Authors:** Germán Rivera Monroy, Renata Murguiondo Pérez, Efraín Weintraub Ben Zión, Oscar Vidal Alcántar-Garibay, Ericka Cristina Loza-López, Emilio Tejerina Marion, Enrique Blancarte Hernández, Lisset Navarro-Torres, Antonio Ibarra

**Affiliations:** 1Centro de Investigación en Ciencias de la Salud (CICSA), FCS, Universidad Anáhuac México, Huixquilucan 52786, Mexico; 2Neuroimmunology Department, Proyecto CAMINA A.C., Ciudad de México 14370, Mexico

**Keywords:** INDP, Alzheimer, Parkinson, ALS, stroke, SCI, TBI, neuroregeneration

## Abstract

Neurodegenerative diseases (NDDs) are a major health problem worldwide. Statistics suggest that in America in 2030 there will be more than 12 million people suffering from a neurodegenerative pathology. Furthermore, the increase in life expectancy enhances the importance of finding new and better therapies for these pathologies. NDDs could be classified into chronic or acute, depending on the time required for the development of clinical symptoms and brain degeneration. Nevertheless, both chronic and acute stages share a common immune and inflammatory pathway in their pathophysiology. Immunization with neural-derived peptides (INDP) is a novel therapy that has been studied during the last decade. By inoculating neural-derived peptides obtained from the central nervous system (CNS), this therapy aims to boost protective autoimmunity, an autoreactive response that leads to a protective phenotype that produces a healing environment and neuroregeneration instead of causing damage. INDP has shown promising findings in studies performed either in vitro, in vivo or even in some pre-clinical trials of different NDDs, standing as a potentially beneficial therapy. In this review, we will describe some of the studies in which the effect of INDP strategies have been explored in different (chronic and acute) neurodegenerative diseases.

## 1. Introduction

The term neurodegenerative disease (NDD) is commonly used when referring to neurological disorders that compromise normal brain physiology, usually related to brain tissue atrophy, neuronal death, reduced mental and cognitive response capacity, which tend to worsen with chronicity [1,2]. This concept embraces typical chronic neural diseases such as Alzheimer’s (AD) and Parkinsonߣs diseases (PD); however, considering that “degeneration” itself refers to the state of going from a basal level of neuronal functioning to a lower one, there is not a specific time lapse required for brain pathologies to become a degenerative disease, thus, acute pathologies such as stroke, spinal cord injury (SCI) or traumatic brain injury (TBI) can be considered as NDDs as well [3].

Despite being chronic or acute, NDDs share a common factor implied in their pathophysiology: neuroinflammation [4]. For many years, neuroinflammation has been a controversial mechanism for scientists due to CNS being considered as an immune-privileged site owing to the presence of the blood–brain barrier (BBB), which prevents the access of peripheral cells and some molecules to brain tissue. Nevertheless, recent research studies have changed this belief by showing an active and constant interaction between the CNS and both the innate and adaptive peripheral immune systems. Novel reports affirm that in healthy individuals, the peripheral immune system and CNS constantly interact in order to maintain CNS integrity, adequate cognitive function and neurogenesis [5].

However, under pathological conditions, as well as by aging, this interaction could be dysregulated and lead to a pro-inflammatory environment, affecting neuron survival and functioning [6].

Likewise, after brain injury, an inflammatory state is induced with the expression of pro- and anti-inflammatory cytokines, which helps to heal and limit the initial injury. Unfortunately, the anti-inflammatory response is usually overcome by a pro-inflammatory phenotype [7,8,9,10,11]. Once activated, microglia secrete potentially harmful cytokines and molecules such as Tumor Necrosis Factor-alpha (TNFα), Interleukin-1 beta (IL-1 β), Interleukin 6 (IL-6), chemoattractants and vascular adhesion proteins. This reaction activates macrophages under an M1 phenotype and produces reactive lymphocyte infiltration, reactive oxygen species (ROS), oxidative stress (OS) and a pro-apoptotic state. Under these circumstances, neurons tend to be damaged as a consequence of ROS and generate a neurotoxic state due to glutamate overproduction and overstimulation of N-methyl-D-aspartate (NMDA) receptor [12,13]. Furthermore, astrocytes suffer from excitotoxicity that leads to an increase in glutamate and calcium ion release. In addition, endothelial cell dysfunction causes impaired nitric-oxide-mediated vasodilation and vascular cells release matrix metalloproteinases (MMPs), generating extracellular matrix degradation and loss of tight junctions’ structure, which together enhance BBB deterioration and leakage [14].

Currently, most NDD therapies are focused on treating symptoms and few of them are able to limit disease progression and subsequent degeneration [15]. Moreover, the increase in life expectancy and NDD incidence turns this group of diseases into an important worldwide concern, not only for health care workers but also for caregivers and all those who are involved in public health services and social welfare [16]. Considering the important role that neuroinflammation plays in NDD pathophysiology, immunization with neural-derived peptides (INDP) stands as a novel therapy for these diseases [17]. The INDP therapeutic approach consists of the inoculation of such peptides to provide beneficial immune functions to treat CNS pathologies. Several studies have focused on demonstrating INDP’s capacity for modulating immune responses from a pro-inflammatory to an anti-inflammatory phenotype, which could lead to inhibition of the initial and secondary injury or chronic progression, lowering damage and increasing life expectancy in NDDs [18].

The aim of this review is to present the most recent information about NDDs and the novel INDP therapy in order to have a better understanding of the action mechanisms and the outcomes that pre-clinical trials have shown in both acute and chronic stages of these pathologies. Likewise, the statements of this review could help to find new therapeutic strategies for each disease, which could give not only the patients but all people involved a better quality of life.

## 2. Neural-Derived Peptides and Protective Autoimmunity

As mentioned above, the immune response was originally considered the main factor that contributes to brain tissue damage and degeneration in NDDs [19]. Currently, research has shown that immune cell infiltration does not represent a harmful event, but it also plays an important role in neuroprotection and regeneration, which was first introduced by Dr. Michal Schwartz; the general paradigm of this proposal was embraced under the concept of “protective autoimmunity” (PA) [20].

PA is considered as a physiological response of the immune system after CNS injury, which activates T cells against neural constituents, leading to the activation of effector cells such as microglia and peripheral macrophages. The aim of this response is to protect and restore the affected area. However, this reaction does not fulfill its purpose, since in the CNS the immune activity is minimal [21].

As stated before, the CNS was considered as an immune-privileged site, and the general belief was that there was not any adaptive immune activity inside. However, Schwartz’s studies showed that the CNS has the capacity to promote a response, which could increase the number of autoreactive T cells that could play a key role in the reduction of neuronal loss. These findings suggested that the CNS is capable of evoking a self-protective T-cell-mediated autoimmune response, however, in order to limit the damage caused by trauma, this response should be boosted [22].

At present, therapies for NDDs based on modulating immune response have been studied, particularly those aiming at the dampening of a harmful pro-inflammatory microenvironment and improving PA. Neural-derived peptides (NDP) are short immunogenic sequences of amino acids, mainly derived from natural CNS components, that have some changes in their structure, meaning the amino acid sequence is modified, which confers on them the capacity to modulate the immune response in order to promote a beneficial microenvironment [23]. This variation in the response leads to T-cell activation into an anti-inflammatory phenotype, enhancing PA. Furthermore, these specific amino acid modifications generate protection against autoimmune reactions, such as encephalomyelitis, since NDP are only partial agonists for TCR and the immune reaction induced by them is under a controlled response [24].

The modulation promoted by NDP has been previously studied, it is recognized that immunization with these peptides is capable of regulating the immune system response against specific neural constituents in neurodegenerative diseases. The change from a Th1 (pro-inflammatory) to Th2 phenotype (anti-inflammatory) induces a neuroprotective and restorative microenvironment with beneficial effects, including the production of neurotrophic factors such as brain-derived neurotrophic factor (BDNF) and growth associated protein 43 (GAP-43). Further, INDP promotes the release of anti-inflammatory cytokines such as Interleukin-10 (IL-10) and IL-4 and upregulation of transforming growth factor beta (TGFβ), dampening the presence of harmful Th1 macrophages [25]. In this scenario, an immune response—in fact, an autoreactive response—can promote beneficial effects based on a protective and restorative microenvironment. The novel therapy with these peptides has also shown the ability to induce neural restoration in acute and chronic stages of the diseases, where the increase in the production of neurotrophic factors is remarkably important for a better recovery [26].

In the past years, several NDPs have been studied for NDD therapy, such as A91, a modified peptide with a specific structure in the amino acid sequence where the lysine residues are exchanged for alanine at position 91 that helps to switch T-cell responses towards a Th2 phenotype. This peptide has been used in experimental models to treat SCI. In these studies, A91 immunization has induced neurogenesis and axonal regeneration. Another well-known NDP is Glatiramer acetate (also known as Cop-1, Copolymer 1 or Copaxone), a synthetic polypeptide constituted by tyrosine, glutamate, alanine, and lysine, which has been used in animal models for cerebral ischemia. It induces the activation of effector T cells, generating modulatory reactions in the immune system [27]. Researchers have reported that Cop-1 could promote Th2 differentiation, leading to neurotrophin type 3 (NT-3), BDNF, TGFβ, neurotrophin type 4 (NT-4) and Interleukin-2 (IL-2) secretion. Lastly, the use of poly-YE, a copolymer conformed by glutamate and tyrosine, has demonstrated its beneficial effects in preclinical studies for stroke. It is believed that poly-YE plays a specific role in the modulation of microglia activity, enhancing the production of neurotrophic factors and boosting T cell response, which produces significant clinical and behavioral improvements in experimental models [28]. All of these novel strategies are promising alternatives for the treatment of NDDs (see Table 1).

## 3. INDP in Chronic Neurodegenerative Diseases

### 3.1. Alzheimer’s Disease

AD is a progressive NDD associated with cognitive impairment and stands as the first cause of dementia around the world. Clinically, patients show certain symptoms, such as the loss of episodic and semantic memory, that relate to neuronal loss from temporal lobes, particularly at the hippocampus level. As the disease progresses, patients become less self-sufficient, have poorer cognition ability and may require assistance with their daily activities [29]. AD pathogenesis involves the presence of two components that play a predominant role in its development: Amyloid-β (Aβ) plaques, an abnormal peptide product of amyloid-β precursor protein misfolding and Tau protein neurofibrillary tangles (NFTs). Under normal physiological conditions, amyloid-β precursor protein moderates cell survival, growth, motility, cholesterol binding and metal ion homeostasis, whereas Tau protein promotes cytoskeleton stability and is involved in vesicle transportation [30]. However, in AD Aβ amyloidogenic fragments oligomerize and form aggregates that eventually accumulate, forming plaques. The formation of these plaques induces the activation of several kinases that, in high concentrations, contribute to Tau hyperphosphorylation, leading to its oligomerization and formation of insoluble NFTs [31].

Neuroinflammation plays an important role in the pathophysiology of AD as well. Deposits of Aβ plaques and NFTs trigger microglial activation and the local inflammatory response. Consequently, macrophages infiltrate to engulf the plaques; these infiltrations produce pro-inflammatory cytokines, OS and ROS [32]. In addition, research suggests that Aβ oligomers induce synapse loss and dysfunction, ion channel alterations, impaired calcium homeostasis, increased mitochondrial OS and diminished energy metabolism. NFTs accumulate within neuronal structures, leading to the loss of neuron intercommunication and cytotoxicity, which together enhance neuroinflammation [33]. With chronicity, neuroinflammation compromises BBB integrity, which predisposes brain tissue to further damage and inhibits neuroprotection mechanisms [34,35]. Currently, acetyl-cholinesterase inhibitors (AChEIs), such as galantamine, rivastigmine, donepezil and memantine (an antagonist of NMDA), are the available therapies more frequently used for AD; however, none of these options are curative or have shown the ability to slow down or stop disease progression [36].

INDP also stands as a potential therapy for AD. Glatiramer acetate (GA), an amino acid copolymer, has proven to substantially limit AD evolution. Diverse preclinical studies in murine models have demonstrated that GA administration (varying between nasal, subcutaneous intravenous, alone or combined with vehicle administration) promotes better outcomes [37]. Initially, in 2005 Frenkel established that nasal administration of GA with a mucosal adjuvant resulted in significant reduction of fibrillar amyloid presence in hippocampal regions that was associated with the reduction of IFN-γ expression in the brain. Studies from 2015 and 2020 demonstrated that weekly subcutaneous GA immunization improved IL-10 levels, Aβ phagocytosis by macrophages, synaptic integrity preservation, astrogliosis restriction and cognitive functions. These results are similar to those obtained by Rentsendorj in 2018, which showed that weekly subcutaneous GA administration promotes the recruitment of peripheral phagocytic cells, induces IL-10 expression and reduces ROS and the pro-inflammatory environment [38,39,40,41].

The application of INDP as a therapeutic target for AD needs further investigation; however, promising results from several studies over the last years set this therapy as a novel alternative for patients in order to stimulate their own immune system to combat neurodegeneration.

### 3.2. Parkinson’s Disease

PD is a progressive, neurodegenerative disorder that was first described by James Parkinson in 1817 [42]. PD is considered as the second most common age-related neurodegenerative pathology, affecting from 5 in 100,000 to more than 35 in 100,000 new cases yearly, with a remarkable increase in its prevalence with age from the sixth to the ninth decades of life, commonly affecting 1% of the population above 60 years [43,44]. The incidence of this disease is higher in males due to estrogen concentrations in females offering profound nerve cell protection [45]. The cardinal symptoms are associated with motor (bradykinesia, rigidity, resting tremors and postural deformities) and non-motor symptoms. PD has a multifactorial etiology resulting from a combination of environmental, genomic and epigenetic factors, although PARK7, leucine-rich repeat kinase (LRRK2), putative kinase 1 (PINK1), Parkin 2 (PARK2) and PTEN-induced, are some of the well-known genes implicated in the development of the recessive and autosomal dominant forms of the disease. A characteristic hallmark of PD is the loss of dopaminergic neurons from the substantia nigra, pars compacta, in the midbrain [46]. Another pathologic feature is the presence of Lewy bodies (LB), which are accumulations of intracytoplasmic concentrations of α-synuclein, a protein that under usual physiological conditions, plays a critical role in vesicle fusion, axonal transport and neurotransmitter release, whereas in PD tends to misfold and aggregate, producing cytotoxic effects [47].

In addition, metabolism of some metal ions, such as copper (Cu) and iron (Fe), have been associated with neurodegenerative conditions, including PD. Cu and Fe are considered redox-active metals, which are essential for optimal brain operations, such as neurotransmitters synthesis, myelin production, synaptic signaling and more, but their high levels produce cell toxicity. However, if their concentrations increase, ROS levels are also going to increase, causing α-synuclein aggregation within LB and lipid peroxidation that lead to DNA destruction and nerve cell degeneration [48].

Neuroinflammatory events have also been postulated as possible contributors to PD pathogenesis, as it has been confirmed in post-mortem studies that many of the apoptotic nigrostriatal dopaminergic neurons showed high amounts of molecules such as IL-6 and TNFα and apoptosis-related factors such as p55, Fas and caspases 1-2 [49]. Furthermore, the intermediary molecules resulting from α-Synuclein accumulation trigger innate as well as the adaptive immunity, promoting microglia activation and ROS production, disturbing synaptic function and causing neuronal degeneration and chronic neuroinflammation. Current treatment for PD is mainly symptomatic, including levodopa, dopamine agonists, catechol-O-methyltransferase inhibitors and monoamine oxidase B inhibitors. Nevertheless, these medicines are focused on PD manifestations rather than limiting disease progression [50]. Likewise, other groups of drugs, such as the AchEIs, memantine and atypical antipsychotics, are focused on treating non-motor symptoms, such as apathy, depression and autonomic dysfunction. Up to the present time, research suggests that INDP should be considered as a treatment for PD due to the beneficial effects that have been reported. Preclinical studies using Cop-1 have demonstrated that after daily administration for 7 days in a murine PD model, mice presented a decrease in midbrain α-Synuclein and microglial marker allograft inflammatory factor 1 (AIF1) levels, an increase in BDNF levels and the animals had an improvement in motor functions, particularly gait and grip strength [51]. Other studies revealed a significant increase of Th2 T cells in mice immunized with Cop-1, generating a neuroprotective environment, increasing astrocyte-associated glial cell derived neurotrophic factor (GDNF) expression and protecting the nigrostriatal system by modulating microglial responses [52]. Further studies have followed the path of GA testing in PD models, obtaining promising results after its periodical administration, such as the increase in anti-inflammatory cytokines such as IL-4 and IL-10 and striatal tyrosine hydroxylase expression and inhibition of dopaminergic cells degeneration, which could delay disease progression [53]. Based on these investigations, INDP remains as a promising therapeutic strategy for PD.

### 3.3. Amyotrophic Lateral Sclerosis

Amyotrophic lateral sclerosis (ALS) is a multifactorial NDD that consists of progressive degeneration and significant death of motor neurons located in the brain and spinal cord [54]. Recent studies reported an incidence between 0.6 and 3.8 per 100,000 person-years; Europe is the continent with a higher prevalence, ranging from 2.1 to 3.8 per 100,000 person years [55]. Clinical manifestations include motor and extra-motor symptoms such as muscle weakness, dysarthria, dysphagia and, in severe cases, total movement restriction and respiratory paralysis [56]. The incidence in America and Europe ranges from one to two cases per 100,000 people yearly. ALS pathogenesis remains partially understood; however, some gene mutations such as superoxide dismutase-1 (SOD1) and TAR DNA-binding protein (TARDBP) appear to be related with the etiology of this condition [57]. In addition, agriculture jobs, exposure to pesticides and heavy metals, smoking and family history of ALS stand as possible risk factors for developing the disease. It is known that a collection of intracytoplasmic protein inclusions inside motor neurons, mainly constituted by the nuclear TAR DNA-binding protein 43 (TDP-43) and in less proportion by SOD1 aggregation, lead to neuronal dysfunction and even death [58].

ALS pathophysiology suggests that the mechanisms mentioned above might be responsible for neuronal damage by sequestering TDP-43 and inhibiting its different physiological functions as well as by promoting a pro-inflammatory and cytotoxic environment in both the brain and spinal cord [59]. Synaptic dysfunction with excessive glutamate expression has been identified as an important issue in its pathophysiology. The high concentrations of glutamate overstimulate action potential firing and produce mitochondrial disfunction, OS, an increase in calcium consumption and storage disbalance, which critically affects neuronal interconnection and integrity. Moreover, synaptic dysfunction, as well as protein inclusions, lead to microglia activation and the release of pro-inflammatory cytokines, such as IL-1β and TNFα, as well as M1 macrophage infiltration [60]. Altogether, these pathological processes generate a chronic neuroinflammatory state that produces neurodegeneration of corticospinal axons, with thinning and scarring of the lateral aspects of the spinal cord. Current therapies for ALS include Riluzole, a glutamate release inhibitor that helps to inactivate voltage-dependent sodium channels and protects neurons against OS, and Nuedexta, a symptomatic treatment for swallowing, speaking and other pseudobulbar affections [61].

INDP has shown to be a promising therapeutic approach for ALS. In the last decade, several studies in murine models showed that immunization with GA increases life expectancy by protecting motor neurons under protective autoimmunity principles. Anti-inflammatory cytokine expression has been observed in daily GA administration for 2 weeks; thereby, results indicate that this therapy increases the presence of motor neurons in comparison to controls, with a consequent improvement in motor activity [62]. Novel immunotherapies for ALS are constantly emerging. For instance, in 2019 a study aimed to test the effectiveness of two vaccines against SOD1 misfolding, was based on the principle that immunizing with disease-specific epitopes can induce specific antibody production and a Th2 immune response. Both vaccines demonstrated their capacity to improve the life expectancy of mice with SOD1 mutation by promoting a Th2 immune response and expression of anti-inflammatory cytokines [63].

INDP, as a therapeutic strategy for ALS, has been successful, and for this reason, stage 1 clinical trials using GA have already been performed, with promising results focused on symptom improvement and patient safety [64]. Nevertheless, further investigation is required to determine if INDP therapy could increase life expectancy and long-lasting outcomes.

### 3.4. Ischemic Stroke

Cerebral ischemia, cerebrovascular ischemia or ischemic stroke (just “stroke” for this paper) is a disease caused by the interruption of blood flow or hypoperfusion that leads to brain tissue injury and, if it is not quickly restored, ischemia [65]. Stroke represents the second cause of sudden death in the world and the most common etiology of neurological disability [66]. According to estimations, there are 16 million first-time stroke occurrences each year, representing 5.7 million deaths. Approximately 17.8% of the population over 45 years have experienced stroke symptoms and silent cerebral infarction occurs in between 6% and 28% of people, increasing with age. Starting from the age of 65, stroke is the leading cause of disability and patients tend to need long-term care and recovery requirements [67].

Clinical manifestations depend on the specific brain area affected by the blood flow imbalance; thus, a front-temporal stroke might manifest with motor and language alterations, whereas visual disturbances or blindness are more characteristic of an occipital one [16]. Stroke survivors commonly present balance difficulties, vision loss, paralysis of the body (specific parts), aphasia, depression and more impairments related to cognitive functions affecting their daily activities [68].

The etiology includes an embolic event, thrombus formation or even vasospasm, as well as risk factors such as smoking, physical inactivity, obesity, diabetes mellitus, hypertension and others [69].

In the past, it was known that a stroke could just be an acute isolated event; however, it is now recognized that stroke produces acute and chronic neurodegeneration that can be classified in two adverse events: the initial and secondary injuries. The initial injury refers to the direct result of acute blood flow interruption, which includes neuronal death and severe injury of neurons in the peripheral zone. The secondary injury is the consequence of microglia activation and the initiation of a neuroinflammatory state. After ischemia, M1 macrophages are recruited, with consequent release of pro-inflammatory mediators such as TNFα, IL-1 β, IL-6 and matrix metalloproteases [70]. In the same way, macrophage infiltration generates ROS and OS which, in addition to metalloprotease production, can directly affect BBB integrity by enhancing endothelial disruption. Together, these reactions generate a cytotoxic and inflammatory environment that, if untreated, will maintain over time and chronically induce cell death and neurodegeneration. Treatment depends on the evolution time and can be classified as intravenous thrombolytics, such as alteplase or tenecteplase, or mechanical thrombolysis, such as endovascular therapy and stent retrievers [71].

Several preclinical studies have evaluated the use of NDP for stroke; GA and its derivates are the peptides more frequently tested. The first investigations, performed in transient middle cerebral artery occlusion (tMCAo) rat models, demonstrated that Cop-1 immunization after reperfusion produces neurological improvement and smaller infarct volume in comparison with controls [72]. These outcomes were supported by further studies that evaluated the influence of Cop-1 administration in the production of neurotrophic factors and anti-inflammatory cytokines in the choroid plexus after cerebral ischemia using the tMCAo model. The result of this study showed that Cop-1 can stimulate the expression of cytokines such as IL-10 and neurotrophic factors such as BDNF, neurotrophin-3 and insulin-like growth factor 1, which together may promote neurogenesis and neuroprotection under PA principles [73]. Controversy exists between the usage of different cerebral ischemia models, as reports using permanent focal cerebral ischemia occlusion (pMCAo) models have found that there is not significant volume reduction after GA administration; however, the M1 microglial downregulation, as well as the increase in anti-inflammatory cytokines production and neurogenesis, persist in both models [74]. Further research is needed in order to find more beneficial outcomes for INDP between different cerebral ischemia models; to date, this therapy has shown solid results, improving neuroprotection and enhancing anti-inflammatory environments, and stands as a promising therapy for stroke.

### 3.5. Spinal Cord Injury

Spinal cord injury (SCI) is a serious and crucial disease that is commonly produced by traumatic mechanisms such as compression, contusion or transection. SCI generates anatomical changes and physiological impairments causing permanent motor and sensory deficits such as spasticity, muscle paralysis, atrophy, gait disorders and pain. Nevertheless, besides muscular, osteoarticular and neuro-psychic systems, this pathology involves many apparatuses of the organism, including the cardiovascular, respiratory, gastrointestinal and genito-urinary systems [75].

According to epidemiological studies, it is estimated that there are 3.6 to 195.4 cases per million people around the world and the most susceptible group is younger people, from the second to the fifth decade of life. In addition to motor or sensitive symptoms, a particular consequence of SCI is chronic neurodegeneration and cognitive impairment. These aspects are usually left aside for attending more evident clinical manifestations, yet neurodegeneration can considerably affect patients’ life quality and prognosis [76].

As well as in stroke, the pathophysiology of SCI is classified as primary or secondary injury. The former is directly induced by a traumatic event or lesion and includes axon disruption and neuronal membrane alteration, as well as damage to blood vessels and nerve connections [77]. Mitochondrial imbalance and a cytotoxic and pro-inflammatory environment are part of the secondary injury, where the immune response is responsible for further damage [78]. Secondary injury is characterized by a pro-inflammatory cascade of events that expands the initial lesion. During this phase, there is microglial activation along with macrophage, neutrophil and T-cell infiltration; moreover, pro-inflammatory cytokines such as TNFα, IL-1α, IL-1 beta and IL-6 are expressed and neuronal physiology and homeostasis is disrupted. There is also an imbalanced calcium consumption and production of ROS and OS [79]. The maintenance of this neurotoxic environment can lead to demyelination and glial scar formation (which sometimes is considered as a third injury phase). Finally, it is believed that neuroinflammation is not limited to the original site of injury and the expression of inflammatory cytokines, as well as metabolic imbalance and neurotoxicity (particularly affecting endoplasmic reticulum), can lead to specific lobar damage and cognitive impairment [80].

Several studies have shown successful results for INDP in treating SCI in preclinical models. The A91 peptide contains an immunogenic sequence of 87–99 amino acids derived from the myelin basic protein. A91 has proven to induce a Th2 response and the expression of an M2 macrophage phenotype after SCI, increasing the production of anti-inflammatory proteins and neurogenesis in the lesion site. Furthermore, A91 has demonstrated to proportionally reduce the apoptotic mechanisms produced after the injury and plays an important role in inhibiting lipid peroxidation [81]. In the long term, A91 has shown multiple benefits such as the improvement of motor performance in rat models, which can be explained by a significant increase in IL-4 and TGFβ, molecules that have been found to convey regenerative processes [82,83]. IL-4 alone promotes proliferation of microglial cells, regulates macrophage responses and inhibits the production of proinflammatory cytokines. On the other hand, TGFβ provides restorative mechanisms, has important effects on adult neurogenesis and participates on neural survival [84].

The GA peptide has also been tested after SCI; however, in contrast with the results obtained in other animal models, GA did not show any beneficial effect [85]. Although more evidence is clearly needed to be applied on the clinical area, A91 is considered as a promising therapeutic strategy for the acute and chronic stages of SCI.

### 3.6. Traumatic Brain Injury

Traumatic brain injury (TBI) is defined as an alteration in brain functions or other evidence of brain pathology caused by an external force applied to the brain. TBI has many etiologies; nevertheless, falls have been identified as the main cause among children and the elderly population, as well as blast trauma for military members. Other common causes are sport and car accidents involving concussion and head trauma [86]. Clinical manifestations may depend on the severity, location and mechanism of the traumatic event, ranging from momentaneous or partial lack of consciousness to even coma or death in severe injuries [87]. It is now recognized that TBI is not just an acute isolated event but a disease with a chronic inflammatory and neurodegenerative component; the damage is classified in two stages. The primary injury consists of the traumatic harm dealt to the brain. The secondary injury is a complex biochemical cascade of events triggered by the primary injury [88].

Neuroinflammation is an essential component for secondary injury and chronic degeneration. Inflammatory-related injury begins after initial neuronal and axonal injury, with the subsequent liberation of damage-associated molecular patterns that activate cell networks, inducing inflammatory gene expression for immune response regulation. Neutrophilic infiltration and the microglial response are triggered in parallel with pro-inflammatory molecules such as TGF1-β, IL-6 and IL 8 production [89]. Recent studies suggest that neuroinflammation enhances glutamate receptor production as well as GABA receptors internalization, which promotes neuronal overstimulation and toxicity. Moreover, cytotoxic and inflammatory environments promote ROS and OS production, which leads to metabolic imbalance, endothelial dysfunction and BBB leakage [90]. Treatment for TBI is limited and includes supportive measures to avoid auditory, visual and physical stress, as well as some interventional therapies such as hyperosmolar therapy, targeting intracranial pressure and hyperbaric oxygen, which has shown apoptosis inhibition, inflammation suppression, BBB protection and angiogenesis/neurogenesis promotion [91].

Despite the fact that there are few studies of INDP specifically targeting TBI, it has been found that intramuscular Cop-1 immunization in a closed head injury experimental model produces a reduction in neuronal loss accompanied with neurological recovery. These improvements might be achieved by a reduction in glutamate production and axonal regeneration [92]. Cop-1 administration stands as a promising treatment for some acute NDDs such as TBI, as it has been found that this peptide might have beneficial functions in improving cognition, even in the absence of disease. According to the study of Nieto-Vera et al., immunizing healthy young rats with Cop-1 enhances cognitive function, particularly learning, and spatial and associative memory; moreover, it increases BDNF production at the hippocampus [93,94]. Further studies are necessary to identify the whole range of beneficial outcomes of INDP therapy in TBI.

## 4. Conclusions

NDDs represent a major health problem worldwide. Current treatments for these pathologies are mainly focused on treating symptoms and they do not have the capacity to limit disease progression or help in CNS functionality recovery. Modulating the immune response has been under research for many years, with the aim of reaching an autoreactive response that would generate a protective phenotype, inducing a physiological state where the immune response is not suppressed but enhanced instead in an anti-inflammatory and self-repairing phenotype. INDP stands as a promising therapy to reach neuroprotection and neurorestoration followed by clinical recovery and disease progression limitation. A variety of NDPs have been studied over the past decades and have shown promising results, including the production of anti-inflammatory cytokines and neurotrophic factors that have led to an improvement in cognition, behavior and motor functions. Further research is needed in order to better understand and define its beneficial properties in specific diseases.

## Figures and Tables

**Table 1 biomedicines-11-00919-t001:** INDP that have been tested for certain NDDs.

INDP Tested in NDDs			
NDD	INDP	Effects	Study Type
AD	Cop-1	Reduction of fibrillar amyloid presence in hippocampal regions, IFN-γ expression in the brain ROS, pro-inflammatory environment. Improvement of IL-10 levels, Aβ phagocytosis, synaptic integrity preservation, astrogliosis restriction and cognitive functions.	Preclinical
PD	Cop-1	Decrease in α-Synuclein and AIF1 levels. Increase in BDNF and GDNF levels, Th2 T cells. Modulation of microglial responses. Improvement of motor functions (gait and grip strength).	Preclinical
ALS	Cop-1	Protection of motor neurons by an anti-inflammatory microenvironment. Increase in the number of motor neurons. Improvement of life expectancy.	Preclinical
	Vaccines against SOD1 misfolding	Induction of specific antibodies. Promotion of a Th2 immune response and expression of anti-inflammatory cytokines. Improvement of life expectancy.	
Stroke	Cop-1	Tested in tMCAo and pMCAo model. Stimulation of cytokines expression such as IL-10, neurotrophic factors such as BDNF, neurotrophin-3 and insulin-like growth factor 1 (tMCAo). Neurological improvement and smaller infarct volume (tMCAo). Downregulation of M1 microglial type (pMCAo). Increase in anti-inflammatory cytokines production and neurogenesis (pMCAo).	Preclinical
SCI	A91	Induction of Th2 response and expression of M2 macrophages phenotype. Increasing in the production of anti-inflammatory proteins (IL-4, TGFβ) and neurogenesis. Reduction of apoptotic mechanisms and lipid peroxidation. Improvement of motor performance.	Preclinical
	Cop-1	Did not show any beneficial effect.	
TBI	Cop-1	Reduction in glutamate production and neuronal loss. Improvement of axonal regeneration and neurological recovery.	Preclinical

AD, Alzheimer’s disease; AIF1, allograft inflammatory factor 1; ALS, amyotrophic lateral sclerosis; BDNF, brain-derived neurotrophic factor; Cop-1, copolymer-1; GDNF, glial cell derived neurotrophic factor; INDP, immunization with neural-derived peptides; IL-4, interleukin-4; IL-10, interleukin-10; NDDs, neurodegenerative diseases; PD, Parkinson’s disease; pMCAo, permanent focal cerebral ischemia occlusion; ROS, reactive oxygen species; SCI, spinal cord injury; SOD1, superoxide dismutase-1; TBI, traumatic brain injury; TGFβ, transforming growth factor beta; tMCAo, transient middle cerebral artery occlusion.

## Data Availability

Not applicable.

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
