# Peer review of "Immunization with Neural-Derived Peptides in Neurodegenerative Diseases: A Narrative Review"

_biomedicines, 2023, doi:10.3390/biomedicines11030919_

Round 1

Reviewer 1 Report

Manuscript #: Biomedicines-2210644

Title:  Immunization with Neural-derived Peptides in Neurodegenerative Diseases: A narrative review

 Authors: Monroy et al

 In this review, Monroy et al, review the current understanding and application of various immunization with neural-derived peptides (INDP) therapy in neurological disease states or in experimental models of neural injury. The authors provide the disease burden and the current peptide-based vaccine intervention approaches in acute and chronic neurodegenerative conditions. The literature has been reasonably well-reviewed, and recent literature has been well-represented. The review covers various types of INDPs currently being used or tested. To improve the scope of the review, additional information/discussion on the following is recommended.

1. Page 2, Line 69-70: Why is this a concern specifically for health care workers only?  What about other workforces and the public in general, wherein exposure to occupational or environmental toxicants are associated with neurological and neurodegenerative disorders?

2. Section 3: INDP in chronic neurodegenerative diseases. The authors should consider including a Table to summarize the various INDPs currently being used, its target, potential action or effect, and the type of study currently used in (clinical or preclinical).  This will be advantageous to a reader to follow, understand, and compare the information easily.

3. The manuscript needs to be reviewed for language, grammar, and spelling accuracy.

Reviewer 2 Report

The present review article assessed the implications of neural-derived peptides in the management of neurodegenerative diseases. The topic is relevant and interesting, with a good contribution to a potential improvement in the management of NDDs. However, a few changes are required in order to improve the present form of the paper. Specific requirements are listed below:

Please check the instructions for authors provided by the journal because the insertion of bibliographic references is incorrect. They should be entered in the MDPI style, which requires the use of square brackets for bibliographic indexes [x], not round brackets (x), and the bibliographic list at the end should contain the information in a predefined order according to the style used by the journal.

In order to follow the rules for abbreviation, AD is the abbreviation for Alzheimer’s disease, and PD for Parkinson’s disease, and they should be used for the first time when they appear in the abstract/main text, and afterwards only the abbreviated form will be used. L35, the first mention in the text, should be associated with the abbreviated form (not in L142). Please revise the abbreviations used throughout the manuscript and correct them.

The information presented is relevant, but the way the information is presented is less useful for an easy understanding/reading of the data presented, as no tables or figures are included. Please reorganize and add some information (pathophysiology, etc.) in the form of tables or figures for more fluid/clear information. For PD, I suggest checking and referring to: PMID: 35562956.

Ca2+ should be written as a power.

The aim of the study is not clearly stated. Which is the novelty of your research, and what contributions are made to the field? Please make the aim of the study more relevant, and insert it as a last, separate paragraph of the Introduction section.

It is important to mention because of the relevance within the central nervous system of copper and iron metabolism. I suggest checking and referring to: PMID: 35054862.

L86, the quotation marks have not closed on protective autoimmunity.

As AD and PD have been more complexly addressed, it is advisable to improve the section on stroke and spinal cord injury (epidemiological data, recovery, neurological outcomes). I suggest checking and referring to: PMID: 36295607 and PMID: 36556286.

Round 2

Reviewer 2 Report

A lot of new red text is bolded in the main manuscript. Please unbold it.

4 new references have been introduced in the references section and they are implied for supporting the authors' responses, but they have not added in the main text as inserted references, in brackets.
